# Advances in Phenazines over the Past Decade: Review of Their Pharmacological Activities, Mechanisms of Action, Biosynthetic Pathways and Synthetic Strategies

**DOI:** 10.3390/md19110610

**Published:** 2021-10-27

**Authors:** Junjie Yan, Weiwei Liu, Jiatong Cai, Yiming Wang, Dahong Li, Huiming Hua, Hao Cao

**Affiliations:** 1Key Laboratory of Structure-Based Drug Design & Discovery, Ministry of Education, School of Traditional Chinese Materia Medica, Shenyang Pharmaceutical University, Shenyang 110016, China; yanjunjue0705@sina.cn (J.Y.); caijiatong1@sina.cn (J.C.); w1292720822@sina.cn (Y.W.); 2Wuya College of Innovation, Shenyang Pharmaceutical University, Shenyang 110016, China; 104040219@syphu.edu.cn; 3School of Life Science and Biopharmaceutics, Shenyang Pharmaceutical University, Shenyang 110016, China

**Keywords:** phenazine, pharmacological activity, mechanism of action, biosynthetic pathway, synthetic strategy

## Abstract

Phenazines are a large group of nitrogen-containing heterocycles, providing diverse chemical structures and various biological activities. Natural phenazines are mainly isolated from marine and terrestrial microorganisms. So far, more than 100 different natural compounds and over 6000 synthetic derivatives have been found and investigated. Many phenazines show great pharmacological activity in various fields, such as antimicrobial, antiparasitic, neuroprotective, insecticidal, anti-inflammatory and anticancer activity. Researchers continued to investigate these compounds and hope to develop them as medicines. Cimmino et al. published a significant review about anticancer activity of phenazines, containing articles from 2000 to 2011. Here, we mainly summarize articles from 2012 to 2021. According to sources of compounds, phenazines were categorized into natural phenazines and synthetic phenazine derivatives in this review. Their pharmacological activities, mechanisms of action, biosynthetic pathways and synthetic strategies were summarized. These may provide guidance for the investigation on phenazines in the future.

## 1. Introduction

Natural products are considered to be especially valuable resources for drug discovery. With the rapid development of technologies for isolation, purification and detection, great interest has been shown in the underexplored natural products. Natural phenazines are mainly discovered from microorganisms of marine and terrestrial. More than 100 natural phenazine derivatives and over 6000 synthetic phenazine derivatives have been investigated so far [1,2,3,4]. Phenazine derivatives are a large group of planar nitrogen-containing heterocyclic compounds and the most important core structure is a pyrazine ring (1,4-diazabenzene) with two annulated benzenes [5,6,7]. Phenazine derivatives differ in their chemical and physical properties based on the type and position of present functional groups. Their oxidation–reduction (redox) and fluorescent properties have attracted increasing attention. Some of them are significant dyes applied in medical and biological industry, while others are developed as efficient fluorescent probes to study the change of biochemical profile in vivo [8,9]. Natural phenazines are produced directly from various microorganisms, including *Pseudomonas* spp., *Streptomyces* spp., *Actinomycete* spp. in terrestrial and marine environments. They, like most other important secondary metabolites, possessed various biological activities and have been extensively studied for a long period of time [10,11]. Phenazines and their derivatives exhibit a broad range of biological activities, such as antimicrobial, antiparasitic, neuroprotective, insecticidal, anti-inflammatory, anticancer activity and so on [12,13,14,15]. Phenazine derivatives could be used as prodrugs due to biological activities, for which pharmacologists and chemists have committed themselves to make them into patent medicines. For example, clofazimine (Figure 1) is successfully applied in clinic as widely used antileprosy and antitubercular drug due to antimicrobial activity and immunosuppressive properties [16]. XR11576, XR5944, NC-182 and NC-190 (Figure 1) belong to fused aryl phenazine derivatives, also they show significant anticancer activity and are under clinical studies [7]. Phenazine derivatives display antibacterial activity mainly against methicillin-resistant due to redox properties [17]. According to reports in recent years, phenazine derivatives possessed antiproliferative activities against various cancer cell lines [18,19,20,21]. Additionally, phenazine derivatives were candidates to be developed as inhibitors of disease-related targets and reported to show activity of inhibition to multiple enzymes [22,23,24,25]. Although phenazine derivatives possessed a broad activity spectrum, the in-depth study was hindered due to the limited resource. Many research groups devoted themselves to carrying out the synthetic work to investigate biological activities of synthetic phenazine derivatives.

Laursen et al. reviewed natural and synthetic phenazine derivatives with regard to biological activities in 2004 [6]. Phenazines and their derivatives had been associated with anticancer activity since 1959. On the basis of significant anticancer activity of phenazines and their derivatives, Cimmino et al. excellently reviewed natural and synthetic phenazines and derivatives about their anticancer activity and mechanisms of action in 2012, covering articles from 2000 to 2011 [26]. In recent years, researchers found various novel structures of natural phenazine derivatives and investigations of pharmacological activity were involved in many aspects. In this review, phenazines isolated from microorganisms, synthetic phenazine derivatives, their pharmacological activities and mechanisms of action were summarized, covering the articles from 2012 to 2021.

## 2. Natural Phenazine Derivatives

In the past decades, according to the published articles, many researchers investigated known phenazines deeply and further evaluated their potent biological activity. Other researchers tried to find novel phenazines from natural sources. Natural phenazines can be categorized according to the types of functional groups and their linking positions on the phenazine core.

### 2.1. Biological Activity of Known Phenazines

Compound **1** (phenazine-1-carboxylic acid, Figure 2) is also called tubermycin B due to its antibiotic activity against *Mycobacterium tuberculosis*. It is widely distributed in various microorganisms as a precursor of many natural phenazine derivatives. Gorantla et al. firstly reported its antifungal activity against major human pathogen, *Trichophyton rubrum*, which could be responsible for causing athlete’s foot, jock itch, ringworm and fingernail fungus infections. The minimum inhibitory concentration (MIC) was 4 mg/mL [27]. Varsha et al. first isolated it from *Lactococcus* BSN307 and investigated its anticancer activity against HeLa cell line (IC_50_ = 20 μg/mL) and MCF-7 cells (IC_50_ = 24 μg/mL). It showed inhibitory activity towards leucine and proline aminopeptidases; thus, it would be used as a potential metalloenzyme inhibitor [28].

Compound **2** (Figure 2) is also a significant phenazine-type metabolite produced by various microorganisms. Cardozo et al. investigated its antibacterial activity against MRSA (Methicillin-resistant *Staphylococcus aureus*) strains and found its synergic effect when combined with silver nanoparticles produced by *Fusarium oxysporum* [29]. Thanabalasingam et al. first isolated it from the leaves of a medicinal plant *Coccinia grandis* [30]. Tupe et al. tested its activity against human pathogen *Candida albicans* (MIC = 32–64 μg/mL), demonstrating its mechanism of antibacterial and antifungal activities via reactive oxygen species (ROS)-mediated apoptotic death; **2** could lead to production of intracellular ROS. ROS caused hyperpolarization of mitochondrial membrane, following externalizing phosphatidylserine, chromatin condensation and DNA fragmentation, thus, inducing apoptosis and, finally, cell death [31]. Kennedy et al. and Ali et al. further investigated anticancer mechanism of **2**. The anticancer activity mechanism was also connected with ROS. p53, Bax and cytochrome C (Cyto-C) were overexpressed while caspase-3 was activated and oncogenic, anti-apoptotic proteins such as poly ADP-ribose polymerase (PARP) and B-cell lymphoma-2 (Bcl-2) family proteins (Bcl-2, Bcl-w and Bcl-xL) were inhibited (Figure 3) [32,33].

Pyocyanin (**7**, Figure 2) is a redox-active phenazine. It is a major virulent factor produced by *Pseudomonas aeruginosa*, which exerts damage effects on mammalian cells. Chai et al. explored pathogenesis of **7** on macrophages. Biological data showed it could promote IL-8 secretion and mRNA expression in a concentration-dependent manner. Signal pathways of the protein kinase C (PKC) and nuclear factor-κ-gene binding (NF-κΒ) were involved in phorbol 12-myristate 13-acetate (PMA)-differentiated U937 cells infected by **7** [34]. Forbes et al. aimed to investigate the pyocyanin role of redox-sensitive mitogen-activated protein kinase (MAPK) by inducing toxicity in A549 cell line. The results showed that pyocyanin-induced cytotoxicity was different from c-Jun N-terminal Kinase (JNK) and p38MAPK signaling pathways. Acute ROS production and subsequent oxidative stress strengthened its toxicity [35]. 1,6-Dihydroxyphenazine 5,10-dioxide (**8**, Figure 2) is also called iodinin. It was discovered to show anti-bacterial activity and weak activity against a mouse tumor model. Sletta et al. firstly isolated it from *Streptosporangium* sp. DSM 45942 from the fjord sediment; **8** showed great antibacterial and antifungal activities against *Candida glabrata* and *Enterococcus faecium*, MIC ranging from 0.35–0.71 μg/mL. Compared with normal rat kidney (NRK) fibroblasts, **8** showed higher selectivity towards leukemia cell line. It was a promising compound to be developed as an anticancer drug, especially those targeting leukemia [36]. Myhren et al. further investigated its anticancer potential against acute myeloid leukemia (AML) and acute promyelocytic leukemia (APL) cells. The results demonstrated its anticancer potency against two selective cancer cell lines and weak toxicity to normal cells. Molecular modeling results suggested that it could intercalate between bases in the DNA, leading to DNA-strand break. The apoptosis progress was associated with Fms-like tyrosine kinase (FLT3) internal tandem duplications, mutated/deficient p53 and activation of caspase-3 [37].

Lee et al. isolated compounds **1**, **3** and **4** (saphenic acid, Figure 2) from a deep-sea sediment-derived yeast-like fungus *Cystobasidium larynigs* IV17-028. These compounds could decrease the production of NO, thus showing inhibitory activity against lipopolysaccharide (LPS)-induced murine macrophage RAW 264.7 cells with EC_50_ values of 17.06 mg/mL (76.1 μM), 14.67 mg/mL (54.7 μM) and 6.15 mg/mL (22.9 μM), respectively [38].

Hifnawy et al. isolated compounds **1**, **5** (phenazine-1,6-dicarboxylate) and **6** (phencomycin) from actinomycetes, *Micromonospora* sp. UR56 and *Actinokineospora* sp. EG49. These compounds demonstrated high to moderate antibacterial and antibiofilm activities against four bacterial strains (*Staphylococcus aureus*, *Bacillus subtilis*, *Escherichia coli* and *Pseudomonas aeruginosa*), with modest cytotoxicity against four cell lines (WI38, HCT116, HepG-2 and MCF-7). They took *Staphylococcus* DNA gyrase-B and pyruvate kinase as targets. Subsequently, in vitro data showed that **1**, **5** and **6** (Figure 2) exerted their bacterial inhibitory activities through inhibiting *Staphylococcus* DNA gyrase-B and pyruvate kinase [39].

### 2.2. Natural Phenazines

#### 2.2.1. Terpenoid Phenazines

Terpenoid phenazines contain common structural feature of isoprenylated C or N side chains and most of them show moderate or weak antibacterial activity. Kondratyuk et al. isolated marine phenazines **9** and **10** (Figure 4) from *Streptomyces* sp. strain CNS284: **9** demonstrated inhibitory activity of NF-*κ*B and cyclooxygenase-2 (COX-2); **10** showed potent (sub-µM) inhibition activity of prostaglandin E2 (PGE2) production. However, these activities monitored did not have a strong correlation with each other. The mechanism of action was not apparent and needed to be further investigated [40]. Ohlendorf et al. isolated geranylphenazinediol (**11**, Figure 4) from a marine sediment *Streptomyces* sp. strain LB173 [41]: **11** bears geranylation at C-4 side. It showed weak antibacterial activity and great inhibitory activity toward human acetylcholinesterase (IC_50_ = 2.62 ± 0.35 μM). Phenaziterpenes A (**12**) and B (**13**, Figure 4) are structurally related to geranylphenazinediol, bearing *O*-geranylation. Song et al. isolated them from *Streptomyces*
*lusitanus* SCSIO LR32. However, compounds **12** and **13** did not show antibacterial activity and cytotoxicity against tumor cell lines [42].

Wu et al. isolated N-prenylated endophenazine **14** (Figure 4) from *Kitasatospora* sp. MBT66. **14** inhibited *B. subtilis* better than positive control drugs Ampicillin and Streptomycin [43]. Han et al. isolated several natural phenazines **15**–**19** (Figure 4) isolated from *Streptomyces* sp. NA04227; **15**–**19** showed moderate inhibitory activity against human acetylcholinesterase and moderate antibacterial activity against *Micrococcus luteus* (MIC = 4 μmol/L) [44].

#### 2.2.2. Glycosylated Phenazines

A few natural glycosylated phenazines have so far been found and reported. The activity of glycosylated phenazines was not remarkable and needed to be further investigated. Rusman et al. firstly isolated deglycosylated phenazines (compounds **20**–**25**, Figure 5) from *Streptomyces* sp. Strain DL-93. None of the compounds exhibited any inhibitory activity against tested bacteria and fungi. However, **21**, **22** and **25** showed weak cytotoxicity against HCT-116 cancer cell line with EC_50_ values of 18 μM, 52 μM and 45 μM, respectively. In vitro biological assay data demonstrated that the weak cytotoxic activity was not associated with DNA intercalations and topoisomerase inhibition. The mechanisms of action were uncertain and needed to be further investigated [45].

Wu et al. also isolated glycosylated endophenazines (compounds **26**–**30**, Figure 5) from *Kitasatospora* sp. MBT66; **26** and **28** contain sugar moiety and the sugar is methylated at 2′-O position. These two compounds are rare in nature and firstly reported; **26**–**30** all showed antibacterial activity to some extent against Gram-positive *B. subtilis*. In addition, **26**–**28** and **30** also showed antimicrobial activity against Gram-negative *E. coli*. Interestingly, they found that glycosylated **27** and **28**, compared with their corresponding aglycone, displayed enhanced activities against Gram-negative *E. coli* [43].

#### 2.2.3. Divergent Fused Phenazines

This class of phenazines contains more than one phenazine-derived moiety. There are a few divergent fused phenazines in nature that have been reported so far. Li et al. isolated diastaphenazine (**31**, Figure 6) from an endophytic *Streptomyces diastaticus* subsp. *ardesiacus*: **31** was a cytotoxic dimeric phenazine, showing antibacterial activity against *S*. *aureus* (MIC = 64 μg/mL). However, **31** was inactive against *E. coli* and *C. albicans* even at 128 μg/mL. Compared with positive control (adriamycin), **31** showed weak cytotoxicity against HCT116, BGC-823, HepG2, HeLa and H460 cell lines with IC_50_ values of 14.9 μM, 28.8 μM, 65.2 μM and 82.5 μM, respectively [46].

Baraphenazines A−C (**32**–**34**, Figure 6) are fused 5-hydroxyquinoxaline/alpha-keto acid amino acid compounds. Baraphenazines D and E (**35** and **36**, Figure 6) are special diastaphenazine-type compounds. In addition, baraphenazines F and G (**37** and **38**, Figure 6) are phenazinolin-type compounds. Wang et al. isolated them from *Streptomyces* sp. PU-10A and investigated their anticancer activity. Only **36** displayed appreciable activity against A549 and PC3 cell lines with IC_50_ values of 2.4 μM and 4.7 μM, respectively. Structure-activity relationship (SAR) indicated that the group of amide on **36** was important to the anticancer activity. On the contrary, the group of free acid on **32**−**35**, **37** and **38** was not benefit to the antiproliferative activity. These bioactivity data could explain a general toxicity-based mechanism of action [47].

#### 2.2.4. Biological Activity of New Phenazines

Kennedy et al. isolated 5-methyl phenazine-1-carboxylic acid (**39**, Figure 7) from a rhizosphere soil bacterium. It showed selective cytotoxicity against A549 and MDA MB-231 cell lines in a dose-dependent manner, with IC_50_ values of 488.7 nM and 458.6 nM, respectively. It exhibited antiproliferative activity by inhibiting cell viability, DNA synthesis and induced G1 cell cycle arrest and apoptosis in cancer cell lines. It was mediated by mitochondrial apoptotic pathway via activation of caspase-3 and down regulation of Bcl-2 expression [32].

Lee et al. isolated compounds **40**–**42** (Figure 7) from yeast-like fungus *Cystobasidium larynigs* IV17-028. These compounds, except for compound **41**, could also inhibit the production of NO, thereby showing inhibitory activity against lipopolysaccharide (LPS)-induced murine macrophage RAW 264.7 cells with EC_50_ values of 18.10 mg/mL (46.8 μM) and 6.15 mg/mL (22.9 μM), respectively [38]. Deng et al. isolated bioactive compound **43** (Figure 7) from *Streptomyces lomondensis* S015, which inhibited *Pythium ultimum*, *Rhizoctonia solani*, *Septoria steviae* and *Fusarium oxysporum* f. sp. *Niveum*, with MIC values of 16 μg/mL, 32 μg/mL, 16 μg/mL and 16 μg/mL, respectively. These biological data showed the potency of **43** as a promising hit for the further development as a biopesticide [48]. Cha et al. isolated compounds **44** and **45** (Figure 7) from *Streptomyces* sp. UT1123. These two compounds had a unique methylamine linker rather than common methyl ether. Additionally, **44** and **45** showed neuronal protective activity on HT-22 mouse hippocampal neuronal cells even in a low concentration [14].

## 3. The Progresses of Biosynthetic Pathways of Phenazines

McDonald et al. found that 2-amino-2-deoxyisochorismic acid could be completely converted into **1**. These compounds were mainly extracted from *Pseudomonas* spp. *PhzB*, *phzD*, *phzE*, *phzF*, *phzG* and so on, which belong to the *phz* gene family and they were proved to play important roles in phenazine synthesis [49]. Chorismic acid (**56**) was not only a common precursor for many primary and secondary metabolism but also the first substrate in biosynthetic pathway towards natural phenazines. Many important phenazines could be produced from microorganisms by this biosynthetic pathway. Combining with previous reports of Xu et al. and Blankenfeldt et al., the classical biosynthetic pathway towards strain-specific phenazines starting from chorismic acid is shown in Figure 8 [5,50]. 

Normally, the biosynthetic pathway in *Pseudomonas* mainly focused on simple modification of phenazine cores. Shi et al. reported a different biosynthetic pathway of various complex phenazines from the entomopathogenic bacterium *Xenorhabdus szentirmaii*. By modifying the core structure of phenazine, such as electron-rich aromatic rings, reduced form nitrogen(s) and carboxylic acid, a variety of natural phenazine derivatives can be generated. The synthesis of compound **59** is controlled by the typical *phz* operon in *X. szentirmaii* similar to classical biosynthetic pathway. Further modification of **59** was diversified by the enzymes from two discrete biosynthetic gene clusters. This progress of biosynthetic pathway was involved in multiple enzymatic and non-enzymatic reactions (Figure 9) [51].

Guo et al. developed a biosynthetic pathway to synthesize phenazine N-oxides in *Pseudomonas chlororaphis* HT66 (Figure 10). They used three enzymes, a monooxygenase (*phzS*), a monooxygenase (*phzO*) and the N-monooxygenase (*naphzNO1*). Additionally, *naphzNO1* only catalyzed the conversion of **80**, but failed to convert into **81** in vitro. This study also provided a promising method for the synthesis of aromatic N-oxides by *naphzNO1* [52].

## 4. Synthetic Phenazine Derivatives

Although natural phenazines possess a variety of biological activities, most of which show moderate or weak activity, thus lacking the possibility to be used as drugs. Structural modification and total synthesis are used to achieve some phenazine derivatives which show notable activity. Normally, the researchers focus on enhancing one special biological activity. Here, synthetic phenazine derivatives will be classified into the following categories in detail, with the perspective of biological activities and functional groups connected to phenazine core.

### 4.1. Antimicrobial Activity

#### 4.1.1. Halogenated Phenazine Derivatives

According to related reports, bacterium would stop growth in MIC of 2–4 µg/mL and die in MIC of 2 µg/mL [48]. Halogenated phenazines derivatives are tested as antibacterial agents which could target multiple persistent bacterial phenotypes effectively and show negligible toxicity against mammalian cells. Antibacterial effect of halogenated phenazine derivatives could be attributed to membrane disruption, interference with redox cascades or electron-flow and the production of ROS [13,53]. Halogenated phenazine derivatives needed to be further developed by chemists due to the great antibacterial activity.

Conda-Sheridan et al. synthesized a series of phenazines derivatives inspired by some natural halogenated phenazines. They found *N*-(methylsulfonyl) amide group in the position of C-4 and halogenated group in the position of C-6 would remarkably improve the activity against MRSA. Compounds **82** and **83** (Figure 11) showed stronger antibacterial activity in these synthetic halogenated phenazines compared to positive drug vancomycin (MIC = 2 µg/mL). The mechanism of action of the most active compound was also investigated, but various tested biological data indicated that **83** did not have correlations with major reported antibacterial mechanisms. The in vitro IC_50_ values of **80** and **81** (Figure 11) against HaCaT cells (immortal keratinocytes) were 118 mM (**82**) and 193 mM (**83**), respectively; **83** seemed to be a promising molecule for the development of MRSA drugs. Additionally, the application of computational methods such as quantitative structure–activity relationship (QSAR) and the prediction of LogP would promote the development of antibacterial drugs [54].

Garrison et al. synthesized a series of phenazines derivatives modified in the positions of C-2, C-4, C-7 and C-8. Compound **84** (Figure 12) proved to be the most potent biofilm-eradicating agent (≥99.9% persister cell killing) against Methicillin-resistant *Staphylococcus aureus* (minimal biofilm eradication concentration (MBEC) < 10 µM), Methicillin-resistant *Staphylococcus epidermidis* (MBEC = 2.35 µM) and vancomycin-resistant *Enterococcus* (MBEC = 0.20 µM) biofilms, while compound **85** (Figure 12) demonstrated antibacterial activity against *M. tuberculosis* (MIC = 3.13 µM) [55]. Yang et al. explored a series of halogenated phenazines derivatives modified in the positions of C-4, C-6 and C-8. They discovered that 6-substituted halogenated phenazines derivatives could enhance biofilm eradication and antibacterial activities against Methicillin-resistant *Staphylococcus aureus*, Methicillin-resistant *Staphylococcus epidermidis* and vancomycin-resistant *Enterococcus*. In addition, Yang et al. synthesized a polyethylene glycol (PEG)-carbonate phenazine derivative **86** (Figure 12). Its water solubility was improved and demonstrated 30- to 100-fold enhancement of antibacterial activities against Methicillin-resistant *Staphylococcus aureus* strains, likely through a prodrug mechanism [53].

Borrero et al. synthesized several halogenated phenazine derivatives which were inspired by natural halogenated phenazine **87** (Figure 13); **87** was selected as a lead antibiotic which displayed great inhibitory activity against *S. aureus* (MIC= 1.56 μM). For example, the activity of **88** increased two folds by systematic structural diversification and the SAR was discussed as shown in Figure 13 [56].

#### 4.1.2. Derivatives of Clofazimine

Although clofazimine (Figure 14) is an antibiotic against multidrug-resistant *M. tuberculosis*, the clinical utility of this agent is limited by its poor physical and chemical properties and the possibility of skin discoloration. TBI-1004 and B4100, modified at different positions (Figure 14), showed stronger anti-*M. tuberculosis* activity than clofazimine. Zhang et al. designed and synthesized a series of riminophenazine derivatives which contained a pyridyl group at the C-3 position of the phenazine core, inspired by previous investigations about the developments of TBI-1004 and B4100. Among these derivatives, compound **97** (Figure 14) demonstrated similar activity against *M. tuberculosis*. Additionally, reduced the possibility of skin discoloration in an experimental mouse infection model as compared to clofazimine. In addition, physicochemical properties and pharmacokinetic profiles of **97** were improved [57]. 

Tonelli et al. also synthesized a series of riminophenazine derivatives which contained quinolizidinylalkyl and pyrrolizidinylethyl moieties. These riminophenazine derivatives were tested against *M. tuberculosis* strains H37Rv and H37Ra, six clinical isolates of *M. avium* and *M. tuberculosis* and three mammalian cell lines (HMEC-1, MT-4 and Vero 76). The best compounds **98**–**101** (Figure 15) showed great inhibition against all strains of *M. tuberculosis* (MIC = 0.82–0.86 μM), **98** showed great inhibition against *M. avium* (MIC = 3.3 μM). The MIC values for clofazimine were 1.06 μM against *M. tuberculosis* and 4.23 μM against *M. avium*, respectively; **98** demonstrated a selectivity index (SI = 5.23) against the human cell line MT-4 comparable with clofazimine (SI = 6.4). Toxicity of **98** against mammalian Vero 76 cell line was quite low (SI = 79) [58].

#### 4.1.3. Derivatives of Phenazine-1-carboxylic acid

Exemplified by compounds **1** and **2**, simple natural phenazines possess great fungicidal activities. Many groups continued to investigate their derivatives, hoping to discovery new eco-friendly agrochemicals. Niu et al. designed and synthesized derivatives of phenazine-1-carboxylic acid (**1**) linking with different amino-acid esters. Compounds **102**–**109** (Figure 16) showed greater activity than **1** (EC_50_ = 66 μg/mL) with EC_50_ values between 5.35 to 8.85 μg/mL. Particularly, **107** (EC_50_ = 6.47 μg/mL) and **108** (EC_50_ = 5.35 μg/mL) showed the best fungicidal activities against *Rhizoctonia solani* Kuhn and none of them had phloem mobility [59].

Taking compound **2** as the lead compound, Zhu et al. designed and synthesized a series of phenazine-1-carboxylic acid (**1**) diamide derivatives. The fungicidal activities were tested by using the inhibitory ratio under 0.2 mmol/L (%) against six phytopathogenic fungi *Rhizoctonia solani*, *Fusarium graminearum*, *Alternaria solani*, *Fusarium oxysporum*, *Sclerotinia sclerotiorum* and *Pyricularia oryzac*. Although all derivatives had fungicidal activities to some degree, the inhibitory activities of most derivatives were lower than control (compound **1**). Compounds **121**–**124** (Figure 1) demonstrated inhibitory rates more than 50% against *R. solani* and *A. solani*. Particularly, **121** showed the most potent fungicidal activity against *R. solani*, with the inhibitory rate of 72.7%. Compound **121** demonstrated the strongest fungicidal activity against *P. oryzae* with the inhibitory rate of 82.0% [60]. 

Han et al. designed and synthesized a series of phenazine-1-carboxylic (compound **1**) piperazine derivatives. Most phenazine-1-carboxylic piperazine derivatives showed fungicidal activities in vitro. Particularly, compound **125** (Figure 17) showed inhibitory activity against all tested pathogenic fungi (*R. solani*, *A. solani*, *F. oxysporum*, *F. graminearum* and *P. oryzac*) with EC_50_ values of 24.6 μM, 42.9 μM, 73.7 μM, 73.8 μM and 34.2 μM, respectively [61]. Lu et al. designed and synthesized the derivatives based on the skeleton of **1**, which contained a series of 1,3,4-oxadiazol-2-yl thioether derivatives. The results of biological assay demonstrated that target compounds possessed moderate to good fungicidal activities against *R. solani*, *S. sclerotioru* and *P. oryzac Cavgra*. Compounds **126** and **127** showed more than 90% inhibitory rate against *S. sclerotioru*. The EC_50_ values of **126** and **127** were 11.16 μM and 30.47 μM, respectively. In addition, the EC_50_ value of **127** against *S. sclerotioru* was 10.49 mM, similar as that of compound **1** [62]. 

Li et al. also designed and synthesized the derivatives of compound **1** containing substituted groups of triazole. Most 3-benzyl mercapto-1,2,4-triazol derivatives demonstrated fungicidal activity against one or multiple plant pathogens in vitro and in vivo. Compounds **137**–**140** (Figure 2) displayed better inhibitory activity against rice blast (*P. oryzae*) than **1**. These results provided valuable references for further studies [63].

#### 4.1.4. Water-Soluble Triazole Phenazine Derivatives

Hayden et al. evaluated water-soluble triazole phenazine derivatives, which were synthesized previously. Compounds **141**–**143** (Figure 18) showed high antimicrobial activity at tested concentrations without cytotoxicity against human epithelial cells and tested biological data suggested that **141**–**143** could interrupt metabolic electron-transfer cascades thereby exhibiting cytotoxicity against *E. coli*, rather than production of ROS [64].

### 4.2. Insecticidal Activity

Podophyllotoxin is a natural product used as the lead compound for the preparation of insecticidal agents. It contains A, B, C, D and E rings. Zhi et al. designed and synthesized a series of podophyllotoxin-based phenazine derivatives modified in the C, D and E rings. In addition, the insecticidal activity of target compounds was investigated which showed insecticidal activity against *Mythimna separata* Walker in vivo. Compounds **148** and **149** (Figure 3) were phenazine derivatives of 4-acyloxypodophyllotoxin modified in the E ring. They demonstrated stronger insecticidal activity than toosendanin [65]. Then, they designed and synthesized a series of oxime derivatives of podophyllotoxin-based phenazines modified in the C, D and E rings. Compounds **153**–**157** (Figure 4) exhibited equal or higher insecticidal activity than toosendanin. The combination of podophyllotoxin and phenazine was proved to enhance insecticidal activity [66].

### 4.3. Antiparasitic Activity

Chagas’ disease, caused by *Trypanosoma cruzi*, is a widely spread endemic disease in American. Alvarez and Minini et al. selected phenazine **158** (Figure 19) from their own chemistry library and investigated its in-depth insight mechanism of inhibition; **158** could bind to a widespread enzyme, triosephosphate isomerase (TIM) from *T. cruzi*. It showed great inhibitory activity against TIM and could be further developed as inhibitors of TIM; **158** showed highly selective inhibition against *T. cruzi* enzyme (TcTIM) and weak inhibition against *T. brucei* (TbTIM), without affecting TIM from *H. sapiens* (HsTIM) and *Leishmania* sp. (LmTIM) [67,68].

### 4.4. Anticancer Activity

#### 4.4.1. Phenazine 5,10-dioxide Derivatives

Phenazine-5,10-dioxide derivatives **1****59**–**1****66** are listed in Table 1. Hernández et al. investigated chemosensitizer effect of compounds **159** and **161** to cisplatin. They showed a significant increase of the antiproliferative activity compared with the control group treated with cisplatin alone, demonstrating sensitization to cisplatin therapy. In addition, **159** combined with cisplatin induced cell cycle arrest on bladder cancer cells, resensitizing the invasive and cisplatin resistant 253 J cell line. It also showed great inhibition activity against histone deacetylase (HDAC) and sensitized chemotherapeutic drugs to better access to DNA, which would cause DNA damage, leading to cell death [69].

Phenazine 5,10-dioxide derivatives have also been reported in development as bioreductive agents. This class of compounds all contain a bioreductive moiety, the N-oxide group and a planar heterocycle moiety, phenazine [70,71]. After the hypoxic selective bioreductive process, the phenazine moiety can interact with DNA causing cytotoxicity in the solid tumour cells [72]. Gonda et al. attempted to find selective hypoxic cytotoxins with additional ability to inhibit DNA topoisomerase II. Inhibitive values in normoxia and hypoxia condition of these compounds were shown in Table 1: **162**–**164** displayed some degree of selectivity; **165** showed non-selectivity towards both conditions, normoxia and hypoxia. Meanwhile, **159** and 1**63** showed the best selectivity; **164** exhibited inhibition of topoisomerase II in hypoxia; **166** showed no inhibition of topoisomerase II in hypoxia and normoxia. The DNA interaction abilities of phenazine 5,10-dioxide derivatives were related to cytotoxicity in normoxia or hypoxia. SAR implicated that the arylethenyl moieties were generally responsible for normoxic cytotoxicity. On the contrary, the group of sulfonamido did not produce selective cytotoxicity whether in normoxia or hypoxia [73].

#### 4.4.2. Benzo[*a*]phenazine Derivatives

According to related reports, benzo[*a*]phenazine derivatives show significant activity of antiproliferation against HL-60 cell line. Topoisomerases, including topoisomerase I and II, have been proved to be effective anticancer targets in drug discovery due to highly over-expression in cancer cells [74,75]. 

Benzo[*a*]phenazine derivatives **167**–**174** are listed in Table 2. Zhuo et al. designed series of benzo[*a*]phenazine derivatives, which alkylated the phenolic hydroxyl group on ring B, introducing different substituted groups on ring D by condensation and followed up with amination. This class of compounds showed good antiproliferative activity. Compound **167** demonstrated good Topo I–DNA cleavage complex stabilizing ability in vivo. Compound **168** showed inhibition of ATPase. Compared with **167**, **168** and **169** were introduced a methoxy group, their inhibitory activity was significantly improved and there was a good correlation between the inhibitory activity and cytotoxic activity. Caspase-3/7 activation assay showed that this class of compounds could induce an apoptotic response in HL-60 cell line [76].

Yao et al. synthesized a series of 7-alkylamino substituted benzo[*a*]phenazine derivatives. Most of these compounds showed better inhibitory activity in HL-60 cell line than the other tested cell lines. The structure–activity relationship studies revealed that the substitution of amino groups on terminal of side chain at N-7 position could improve the Topo I/II inhibitory activity and cytotoxicity: **170**–**172** with the dimethylamino terminal showed good Topo I and Topo II inhibitory activity; **170** could stabilize the Topo I-DNA cleavage complexes in vivo; **173** with methoxy group at position C-9 exhibited good Topo I and Topo II inhibitory activity at 25 mM concentration; **173** showed inhibition of ATPase [22].

#### 4.4.3. Pyran[2,3-*c*]phenazine Derivatives

Phenazine derivatives and pyran derivatives are important heterocyclic compounds which possess good biological activity. The heterocyclic pyran structure usually is a functional framework which appears in amounts of important drugs and natural products. Molecular hybridization strategy shows great prospect in the present drug discovery to reduce the side effects and the occurrence of drug-resistance [77]. The aromatic interlayer coupling structure of phenazine, as well as the structural characteristics of fluted or specific enzyme binders, leads to selective high-affinity binders that target DNA and DNA-enzyme complexes [78]. According to recent articles, pyran[2,3-*c*] phenazine derivatives mainly showed cytotoxicity against HepG2 cell line. In addition, the mechanism inducing apoptosis against cancer cells about this class of compounds is shown in Figure 3.

Lu et al. designed and synthesized a series of pyran[3,2-*a*] phenazine derivatives. Most pyran[3,2-*a*] phenazine derivatives demonstrated cytotoxicity against HCT116, MCF-7, HepG2 and A549 cancer cell lines in vitro. Especially, compounds **181**–**183** (Figure 5) were found to show potent growth inhibitory activity against HepG2 cell line (IC_50_ of 2–6 μM). In addition, they also used experimental mouse models to test in vivo activity of these phenazine derivatives. Among these phenazine derivatives, **181** was selected to do tumor xenografts experiment to test the effect of inhibition. H22 cells was injected into ICR mice, inhibitions were 7.78% (5 mg/kg), 68.89% (10 mg/kg) and 77.78% (20 mg/kg), respectively. Further mechanism studies implicated **181** acted as topoisomerases I and II dual inhibitor, cell cycle arrester and apoptosis inducer against HepG2 cell line [23].

Additionally, Liao et al. firstly found **181** as thioredoxin reductase I (TrxR1) inhibitor against HepG2 cell line. TrxR1 is a novel anticancer target different from topoisomerases I and II. Molecular docking was carried out to study the inhibitory possibility of compound **181** against TrxR1. Soon afterwards, they investigated the crucial downstream protein TrxR1 to confirm antiproliferation function of **181** against the HepG2 cell line. It supplied valuable information for further development of the TrxR1 inhibitors [24].

Lu et al. also designed phenazine derivatives containing phenazine, pyran, indole and 1,2,3-triazole pharmacophores. Among these derivatives, **187** (Figure 6) demonstrated the best antiproliferative activity against A549 cancer cell line (IC_50_ of 5.4 μM) and no cytotoxicity against L02 and HUVEC non-cancer cell lines [18].

#### 4.4.4. Benzo[*a*]pyran[2,3-*c*]phenazine Derivatives

This class of phenazine derivatives, which contain phenazine, pyran and benzo core, possess great anticancer activity but rarely reported. Inspired by **XR11576** (Figure 1), benzophenazine derivative has been proved to be a great antitumor compound which could be further modified. Gao et al. synthesized a series of benzo[*a*]pyran[2,3-*c*] phenazine derivatives and evaluated their biological activities. Some of them (compounds **188**–**191**) are listed in Figure 20. Among these compounds, **189** and **191** possessed a *p*-dimethylamino substituted group in benzo ring and the apparent differences between their structures possessed cyan group and ethyl acetate group, respectively, in 3-position of *γ*-pyran ring. Compared to the positive control drug (hydroxycamptothecin), **189** showed better inhibition activity against tested cancer cell line HepG2. However, **191** demonstrated no inhibitory activity against all selective cell lines. These experimental data proved that cyan group played an important role in antiproliferative activity: **189** and **190**, possessing a *p*-dimethylamino group and *p*-hydroxyl group, respectively, demonstrated some cytotoxic activity to HCT116 cell line, IC_50_ of 0.22 μM and 15.32 μM, respectively; **189** showed cytotoxic activity against HepG2 cell line with IC_50_ of 6.71 μM; **188**–**191** showed weak or no activity against MCF-7 and A549 cell lines. SAR studies showed that cyan group and *p*-dimethylamino group played a significant role in antiproliferative activity and resulted in the decrease of cytotoxic activity [79].

#### 4.4.5. Benzo[*a*]chromeno[2,3-*c*] phenazine Derivatives

Chromenes are also an important class of heterocyclic compounds which exhibit attractive pharmacological properties, such as antitumor, anti-vascular, antioxidant, antimicrobial, sex pheromone, tumor necrosis factor-α (TNF-α) inhibitor, cancer therapy and central neuroprotective activities [80]. Chromenes and phenazine derivatives both have attracted attention, but benzo[*a*]chromeno[2,3-*c*] phenazine derivatives have rarely been investigated [81].

Reddy et al. synthesized a series of phenazine-chromene hybrid molecules by protocol of one pot multi-component reaction. Compounds **192**–**194** (Figure 21) showed excellent in vitro antioxidant activity which was benefit to the treatment of cancer and prevention of cardiovascular diseases. Compared with anticancer drugs etoposide and camptothecin, **192**–**194** showed similar antiproliferative activity against selective cancer cell lines with IC_50_ values of 3.28 μM, 4.31 μM and 4.01 μM against HeLa cell line and 2.24 μM, 3.81 μM and 3.12 μM against SK-BR-3 cell line, respectively. These compounds did not show significant toxicity against normal human breast cells (HBL 100) at lower concentrations [19].

#### 4.4.6. Derivatives Derived from 2-Phenazinamine

Inspired by previous reports, Gao et al. isolated a natural derivative derived from 2-phenazinamine, showing high cytotoxicity against selective cancer cell lines. A series of 2-phenazinamine derivatives, including compounds **195**–**202** (Figure 22), were synthesized. The antiproliferative activity results showed that IC_50_ values of **195** and **197** against HepG2 cell line were 16.46 μM and 15.21 μM, respectively. IC_50_ values of **195** and **198** against K562 cell line were 33.43 μM and 49.20 μM, respectively. In addition, IC_50_ value of **196** against MCF-7 cell line was 11.63 μM. Notably, these active compounds showed no cytotoxicity on the epithelial cells from the 293T non-cancer cells. Moreover, mechanism of **195** was similar to control drug (cisplatin), inhibiting cancer proliferation by inducing apoptosis [82].

According to the report of Gao et al., Kale et al. selected some 2-phenazinamines derivatives and further utilized computational methods to investigate their protein targets. The experimental data of **199**–**202** showed great binding energy against BCR-ABL tyrosine kinase by Autodock 4.2. Scores of **199**–**202** were −7.6 kcal/mol, −8.8 kcal/mol, −7.2 kcal/mol and −7.1 kcal/mol, respectively. In addition, the score of imatinib was −8.7 kcal/mol. All the computational data showed 2-phenazinamine derivatives would be inhibitors against BCR-ABL tyrosine kinase and needed to be further investigated [25].

#### 4.4.7. Derivatives Derived from 2,3-Diaminophenazine

Protein kinases (PKs) are essential in many cellular processes, which catalyze phosphorylation of different cellular substrates. Then, phosphorylation in turn regulates various cellular functions. Normally, their activity is strictly regulated. Under pathological conditions, PKs can be deregulated, leading to changes in the phosphorylation, resulting in uncontrolled cell division, inhibiting apoptosis and other abnormalities. Various cancers are known to be caused or accompanied by deregulation of the phosphorylation. Screening new potent, selective and less toxic compounds to inhibit PKs has been proved to be a promising cancer treatment strategy [83].

Mahran et al. prepared some phenazine derivatives, such as compound **203** and its analogues (Figure 23). Among these derivatives, compounds **203**–**207** displayed antiproliferative activity with IC_50_ values of 8.8 μM, 7.7 μM, 8.4 μM, 6.8 μM and 8.8 μM against A549 cell line and 6.0 μM, 4.4 μM, 5.2 μM, 16 μM and 6.8 μM against HCT116 cell line, respectively. These compounds also showed inhibitory activity of human tyrosine kinases. The experimental results of inhibiting human tyrosine kinase were consistent with cell cytotoxicity activity [20].

#### 4.4.8. 2,3-Dialkoxyphenazine Derivatives

Endo et al. reported the only antitumor activity of 2,3-disubstituted phenazines against sarcoma and carcinoma tumors in 1965 [84]. The presence of long fatty chains could provide a good effect to cross the lipid barrier [85]. Moris et al. synthesized 2,3-dialkoxyphenazine derivatives using an easy, efficient and straightforward condensation method. These compounds **208**–**211** (Figure 24) were firstly reported to show activity on MiaPaca pancreatic cell lines with IC_50_ of 0.06 μM, 21 μM, 75 μM and 7 μM, respectively. Interestingly, **208** and **209** interacted with DNA through hydrogen bonds remarkably, showing significant anticancer activity. Compared to Gemzar^®^, **208** and **211** were the most effective ones against pancreatic MiaPaca cell resistant lines. The experimental results showed that the carboxyl substituents on position 7 did not interact with each other through hydrogen bonds. Although possessing planar structures, these derivatives did not have similar mechanism of action as Gemcitabine. In vivo study on mice, **211** was as efficient as Gemzar^®^ at a ten times lower concentration (1 mg/kg vs. 10 mg/kg) [21].

## 5. Conclusions

In this review, we introduced natural phenazines and synthetic phenazine derivatives, which were reported from 2012 to 2021. The biosynthetic pathways of natural phenazines, sources of microorganism and operon genes were illustrated in detail. Additionally, their pharmacological activities, mechanisms of action and structure–activity relationships were also summarized. In future studies, first of all, it is still necessary to find novel structures from natural sources for the screening of lead compounds. Secondly, it is very important to design and synthesize new compounds based on existing SAR through structural modification. The successfully applied drug clofazimine and XR11576, XR5944, NC-182 and NC-190 in clinical studies are good references. The structure modification at C-2, C-3, C-4 and N-6 sites and ring fused derivatives have a good prospect. In addition, target-based drug design can reduce the randomness. Finally, more extensive activity screening will enable more efficient use of compound resources.

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
