# Peer review of "Advances in Phenazines over the Past Decade: Review of Their Pharmacological Activities, Mechanisms of Action, Biosynthetic Pathways and Synthetic Strategies"

_marinedrugs, 2021, doi:10.3390/md19110610_

Round 1

Author Response

This review article summarize the biological activities of natural and synthetic phenazine derivatives reported in the articles published from 2012 to 2021. The reviewers have identified several incorrect structures and I noted additional problematic structures in my review - please check those carefully for accuracy. The weakness of the article can be clearly seen in the last sentences of this section, vague sentences without any clear statement and without any clear message to the reader, I suggest the following modifications:

Point a: The structure of clofaminine was not correct. Please replace the sec-butyl with an isopropyl moiety.

Response a: Thank you for all the helpful suggestions. We have corrected the structure of clofaminine.

Point b: The structure of XR11576, NC-190 and NC-182 can be summarized with a general structure.

Response b: We have summarized a general structure of XR11576, NC-190 and NC-182.

Point c: Line 118. Please replace “1,6-dihydroxy-phenazine 5,10-dioxide” with “1,6-Dihydroxy-phenazine 5,10-dioxide”.

Response c: “1,6-dihydroxy-phenazine 5,10-dioxide” has been replaced with “1,6-Dihydroxy-phenazine 5,10-dioxide”. All the changes in the revised manuscript are colored in red.

Point d: The order of presentation should be: compounds 1,2,5,6,7,8, 3 and 4.

Response d: The order of compounds in the revised manuscript has been corrected carefully.

Point e: Glycosylated phenazine: the exact linking between sugar and phenazine was not clear. Which position of the sugar was involved in the linking with the phenazine ring?

Response e: We have corrected the structures in Figure 5 to show the linking position clearly.

Point f2.2.4. Other bioactive natural phenazines. This section can included in the chapter “2.1. Biological activity of known phenazines”.

Response f: The title “Other bioactive natural phenazines” has been corrected to “Biological activity of new phenazines”, which contains novel compounds never reported before 2012.

Point g: Figure 8, Compounds 59 and 1. Please take a look to the carbon hybridazion (avoid vedged bond in sp2 carbon!).

Response g: We have corrected the stucture of compounds 59 and 1 (remove vedged bond in sp2 carbon).

Point h: Conclusion section. I suggest to focus this section on the most important molecules rather than being comprehensive, bringing out key drug design aspects that are informative.

Response h: We have corrected the conclusion section and added related content about these important molecules.

Reviewer 2 Report

This review article manuscript is not well written and will require extensive English and technical terms checkup prior to the acceptance recommendation to publish. The manusciprt only describe the reseach during the past 10 years ; however, the title of this manuscript might be slightly misleading. There are lots of typos and poor word choices used in this manuscript.

For example:

  • Lots of sentences begin with the word 'And' which is not normally used in academic writing. A suitable conjunction might be more appropriate.
  • Scientific name are not written in italic for many microbial names.
  • Many abbreviation in this paper is used without prior introduction to the reader. The author may need to add a list of used abbreviations somewhere in the manuscript.
  • The word 'biosynthesis pathway' should be replaced with 'biosynthetic pathway.
  • The title of the manuscript should be rewritten to reflect the content of the manuscript. 'Advances in Phenazines over the Past Decade: Review of Their Pharmacological Activities, Mechanisms of Action, Biosynthetic Pathway and Synthetic Strategies'

Ane some other specific suggestions:

  • Line 11 bioactivities should be replaced with biological activities.
  • Line 15 activity should be replaced with activities and And amount of researchers with Many researchers.
  • Line 19 clarifies - categorised
  • Line 20 bioactive- biological
  • Line 37 attention, some - attention. Some
  • Line 38 biology, the - biology, while the 
  • Line 42-43 theres not a good evidence to claim that phenazines has been studied for the longest period.
  • Line 48 define the word 'popular'. other words may be needed here.
  • Figure 1 should show the general phenazine structure with numbering carbon which will be used throughtout the manuscript as this numbered carbon is used frequently.
  • Theres no need to highlight phenazine structure in colour all the time. It's quite disturbing.
  • Line 74 have contributed
  •  Line 75 to investigate 
  • Line 77 sources. Natural phnazine can be categorised
  • Line 82 to its antibiotic
  • Line 96 define the word 'popular'. 
  • Line 96 tested its activity
  • Line 98 activities via
  • Line 108 'about' replace with 'of'
  • Line 109 apoptotic pathway
  • Line 110 virulent
  • Line 112 explored pathogenesis of 3
  • Line 113 , and
  • LIne 117 , and
  • LIne 133 , and
  • Line 135 , with EC50
  • Line 149 chains, and most
  • Line 155 which carbon are you referring to?
  • Line 170 remove 'There are'
  • Line 177 'had nothing to do' - was not associated
  • Line 186-189 they found that glycosylated 27 and 28, compared with ... aglycone, displayed enhanced activities against....
  • Line 192 nature that have
  • Line 207 Mm??
  • Line 209 did not work??
  • Line 237 family and they were proved
  • Line 239 metabolitesis?? metabolism might be the correct term.
  • Line 252-254 poor sentence stucture needed to be rewritten
  • LIne 257 pathway in X.
  • Line 276 Are you referring to bacteriostatic and bacteriocidal activities? A better term can be used here. 
  • Line 285 Where are C-4 and C-6?
  • Line 314 discussed as shown in Figure 13
  • I dont think 'Clofazimine' need to be capitalised. 'clofazimine' should be used instead
  • Line 327-329 poor sentence stucture needed to be rewritten
  • Line 346 Exemplified by compound
  • Line 352 , and 
  • LIne 365 , and
  • Line 396 , and the tested 
  • Line 421 Chagas' disease, caused by.........., is a widely....

Author Response

This review article manuscript is not well written and will require extensive English and technical terms checkup prior to the acceptance recommendation to publish. The manusciprt only describe the reseach during the past 10 years; however, the title of this manuscript might be slightly misleading. There are lots of typos and poor word choices used in this manuscript.

Point 1: Lots of sentences begin with the word 'And' which is not normally used in academic writing. A suitable conjunction might be more appropriate.

Response 1: Thank you for all the helpful suggestions. We have polished the whole manuscript and all the changes are colored in red.

Point 2: Scientific name are not written in italic for many microbial names.

Response 2: The microbial names have been corrected to italic.

Point 3: Many abbreviation in this paper is used without prior introduction to the reader. The author may need to add a list of used abbreviations somewhere in the manuscript.

Response 3: We have added the full name of all the abbreviations in the manuscript to make them clear to the readers.

Point 4: The word 'biosynthesis pathway' should be replaced with 'biosynthetic pathway'.

Response 4: We have replaced 'biosynthesis pathway' with 'biosynthetic pathway'.

Point 5: The title of the manuscript should be rewritten to reflect the content of the manuscript. 'Advances in Phenazines over the Past Decade: Review of Their Pharmacological Activities, Mechanisms of Action, Biosynthetic Pathway and Synthetic Strategies'.

Response 5: We have corrected the title as the suggestion to better reflect the content of the manuscript.

Point 6: Ane some other specific suggestions: …. Line 421 Chagas' disease, caused by.........., is a widely....

Response 6: We have corrected them point by point and polished the whole manuscript carefully. Thanks again for all the helpful suggestions.

Reviewer 3 Report

The authors submitted an interesting review, which deals with pharmacological activities of phenazine derivatives. The topic of this paper is important and timely because phenazine derivatives show many bioactivities such as antibiotic, anticancer and others.

The authors wrote comprehensive review, which includes all important aspects of this topic and cites the most important literature. The text has a logic structure. Furthermore, the authors pointed out many effect-chemical structure relations by illustrative figures. In general, this paper proves high professionality of the author team. Since I found no serious weaknesses, I recommend the manuscript for publication.

Author Response

Thanks a lot for the review of the manuscript and recommendation.

Round 2

Reviewer 1 Report

The authors have properly answered to all my comments. I suggest to accept the manuscript in the present form. 

Author Response

Thanks a lot for  the review of our manuscript.

Reviewer 2 Report

I would like to thank the authors for the updated version of the manuscript; however, there are still a few minor typos that I would like to mention.

Line 85 Mycobacterium 

Line 362 Exemplified

I would like to kindly suggest that the authors should proofread the manuscript thoroughly or using a proper proofreading service priot to the acceptance. 

Author Response

Thank you very much for the suggestions of the two rounds of review. There were indeed some clerical errors in the revised manuscript. We have thoroughly revised the manuscript and marked all the revisions in red.